# VEP Score of a Left Eye Had Predictive Values for Achieving NEDA-3 over Ten Years in Patients with Multiple Sclerosis

**DOI:** 10.3390/s22228849

**Published:** 2022-11-16

**Authors:** Svetlana Miletic-Drakulic, Ivana Miloradovic, Vladimir Jankovic, Ana Azanjac-Arsic, Snezana Lazarevic

**Affiliations:** Department of Neurology, Faculty of Medical Sciences, University of Kragujevac, Svetozara Markovica 69, 34000 Kragujevac, Serbia

**Keywords:** multiple sclerosis, visual evoked potentials, optic neuritis, NEDA

## Abstract

Background: The aim of this study was to determine the predictive value of visual evoked potentials (VEPs) in patients with relapsing–remitting multiple sclerosis (RRMS) in achieving no evidence of disease activity-3 (NEDA-3) during up to 10 years of first-line immunomodulatory therapy and to determine whether the lateralization of optic nerve damage may have prognostic significance concerning clinical disability and response to therapy.Methods: In a retrospective study, a total of 83 patients (53 female and 30 male) with RRMS participated. The average age of patients was 38.31 ± 9.01. Patients were followed for 2, 5 or 10 years. VEPs were measured at the beginning of the follow-up and after many years of monitoring. Data on optical neuritis (ON) were obtained from medical history. The degree of disability was estimated by the neurologist (independent rater), and magnetic resonance (MR) imaging of the endocranium was performed with gadolinium contrasts. Achieving NEDA-3 is considered a favorable outcome of treatments.Results: Among those treated, 19 (22.9%) reached NEDA-3, while 64 (77.1%) did not reach NEDA-3. The values of the evoked potential (EP) score for the left eye (r = 0.008, odds ratio (OR) = 0.344 (0.156–0.757)) and latency for the left eye (r = 0.042, OR = 0.966 (0.934–0.999)) at the onset of disease were predictive factors for achieving NEDA-3. Conclusions: A normal VEP at the beginning of RRMS increases the chance of reaching NEDA-3 by about six times.

## 1. Introduction

Optic neuritis (ON) occurs as the first symptom in 20% to 40% of patients with multiple sclerosis (MS), while about 50% to 70% of such patients have an episode of ON within subsequent relapses [1,2,3,4]. ON is an inflammatory lesion of the optic nerve that can lead to a partial or complete loss of vision [5]. Acute vision loss results from the action of inflammatory mediators, and rapid recovery occurs after their elimination [6]. Possible mechanisms involved in long-term recovery may be the plasticity of sodium ion channels along the demyelinated portion of the axon [7], the reorganization of cortical optical fields [8] or remyelination. During recovery after an ON episode, two opposite processes are present in the first 3 years. On the one hand, the reparative remyelination process may dominate (ion channel reorganization remains a possible alternative), leading to recovery. This plays an important role in protecting axons from degeneration and demyelination [9]. On the other hand, it has been shown that axonal degeneration, which is responsible for disease progression, occurs very early and is also present in active lesions.

There are numerous studies indicating that optic nerve lesions are closely correlated with the atrophy of white and gray matter in the brain; thus, ON lesions reflect the pathological mechanisms that affect the central nervous system (CNS) in this disease and lead to axon loss [10,11].

Visual evoked potentials (VEP) are an important tool for the evaluation of patients with clinical and subclinical ON [12]. VEP characteristics such as prolonged cortical response latencies indicate demyelination processes, while reduced wave amplitude reflects axonal lesions and retinal ganglion cell apoptosis. The presence of abnormal EP during the first episode of ON is a factor predicting the clinical conversions to MS that is more sensitive than the initial magnetic resonance (MR) finding [13].

In recent years, progress has been made in the treatment of MS by using therapy that modifies the course of the disease. Currently, the attitude is to start treatments as soon as possible and thus prevent the development of disability [14]. A new strategy has emerged in recent years—“treating to target”—and it aims to eliminate disease activity or NEDA (no evidence of disease activity) [14,15]. The three components that make up NEDA-3 are the absence of relapse, no disability progression and the absence of disease activity on endocranial MR images [15].

The aims of the study are to determine the predictive value of VEP in patients with relapsing–remitting multiple sclerosis (RRMS) in achieving NEDA-3 after up to 10 years of first-line immunomodulatory therapy and also to determine whether the lateralization of optic nerve damage has any prognostic significance for clinical disability and good response to therapy.

## 2. Materials and Methods

This research involved 83 subjects with RRMS conducted at the Clinic of Neurology Clinical Centre of Kragujevac. Respondents signed a Good Clinical Practice (GCP) form before any of the study’s procedure.

The participants had a clinically definite relapsing–remitting form of multiple sclerosis (RRMS) in whom the diagnosis of the disease was made on the basis of McDonald’s 2017 criteria [16]. Clinical evaluation and VEP were performed at specific time points: at the beginning of follow-up and then after 2, 5 or 10 years from the beginning of therapy. The excluding criteria were as follows: a history of systemic corticosteroids (oral or intravenous) in the previous 6 months; other immunomodulatory and immunosuppressive therapy also in the previous 6 months; the existence of other CNS diseases. All respondents were right-handed.

On the basis of medical documentation, data were collected on the duration of the disease, its course, the number and frequency of relapses, therapy used and the length of treatment. The degree of clinical disability was determined using the Kurtzke expanded disability status scale (EDSS) [17].

All patients underwent VEP (p100), and the amplitude and latency of the waves were measured. Standard clinical testing involves visual evoked potential recordings in which the visual stimulus was a black-and-white checkerboard that is patterned, has 1 Hz alternation and does not have filters. During recording, the black and white fields occasionally change places. VEPs represent the average response to these changes. Responses are recorded using three electrodes covering the occipital region, with one frontal electrode for comparison. The signal from the middle occipital electrode normally contains a prominent positive component that occurs approximately 100 ms after the stimulus and is called the P100. It is preceded by a smaller negative component with a latency of about 75 ms and is called N75. Waveforms at the lateral electrodes are quite variable; thus, the P100 latency obtained at the middle electrode is taken as a measure of retino-striatal conduction. Cerebral responses are amplified and averaged by a computer. VEPs were generally elicited by the monocular stimulation of each eye. The patient’s distance from the stimulation monitor was 1 m in a sitting position.

The findings were scored from 0 to 4 when both eyes were taken into account, where 0 represents normality, 1 represents extended latencies and 2 represents extended latencies and reduced amplitudes in each eye separately [18]. Magnetic resonance imaging (MRI) of the endocranium was performed, and the number of gadolinium-positive lesions was monitored [19]. Patients in whom there was no relapse, no progression of disability and no endocranial MR lesions were found to have no disease activity [14,20]. Namely, they achieved the goal of immunomodulatory therapy—NEDA-3.

The collected data were processed by obtaining the arithmetical mean, standard deviation, statistically median, quartiles, frequencies and percentages. The normality of the distribution of numerical variables was checked using the Shapiro–Wilk and Kolmogorov–Smirnov tests. The correlation of category variables was examined using the chi-square test for contingency tables and of numerical variables using Spearman’s correlation coefficient. Student’s t-test for independent samples and the Mann–Whitney test were employed to compare the mean values of variables for the two populations, while the analysis of variance and Kruskal–Wallis tests were used for comparing the mean values of variables from several populations. The dependence of the binary variable on the other variables was checked using univariate and multivariate binary logistic regression. The analysis was performed in the SPSS statistical package, version 23.0 (IBM corporation, Armonk, NY, USA).

## 3. Results

The 83 patients who participated in the study were observed for up to 10 years after initiating interferon beta and glatiramer acetate therapy. All were diagnosed with definitive RRMS according to the revised McDonald criteria [16]. We did not include patients who were lost during the ten-year follow-up period. Among the participants, 53 (63.9%) were women and 30 (36.1%) were men. 

The distribution of patients according to the time period of follow-up is shown in Table 1. There were no statistically significant differences in the number of patients by gender during the corresponding time periods (*p* = 0.385).

Table 2 shows the demographic and clinical characteristics of patients in the groups monitored for 10 years, 5 years and 2 years. Differences in mean age, disease duration, treatment duration, and disease progression index between different follow-up time intervals were statistically significant (r < 0.0005).

A total of 19 (22.9%) patients reached NEDA-3, while 64 (77.1%) did not. The score for the left eye obtained at the time of diagnosis was associated with a favorable outcome with respect to the treatment of the disease (reaching NEDA-3) (r = 0.024).

Among patients with a normal VEP finding (score 0), 15 out of 40 (37.5%) reached NEDA-3, while 2 out of 18 patients (11.1%) with prolonged latency (score 1) reached NEDA-3, and 2 out of 23 patients (8.71%) with prolonged latency and a reduced amplitude (score 2) reached NEDA-3. Both patients (100%) who had unformed waves (score 3) failed to reach NEDA-3. 

The distribution of the achievement of NEDA-3 after 2-year, 5-year and 10-year monitoring is shown in Table 3.

The difference between the mean latency values obtained in the eye affected by ON between patients who achieved NEDA-3 and those who did not was not statistically significant (r = 0.694). The difference in mean amplitudes obtained in the eye affected by ON between patients who achieved NEDA-3 and patients who did not (r = 0.681) was also not statistically significant.

The factors for achieving NEDA-3 and factors influencing increases in EDSS using binary logistic regression (*p* < 0.05) are shown in Table 4. 

The score on the left eye obtained at the time of diagnosis affected the achievement of NEDA-3 (r = 0.008, OR = 0.344 (0.156–0.757)). The higher the score on the left eye, the lower the chance of attaining NEDA-3. A score higher by 1 reduced the chance of reaching NEDA-3 by about three-fold. The latency value in the left eye influenced the achievement of NEDA-3 (r = 0.042, OR = 0.966 (0.934–0.999)). The higher the latency value in the left eye, the lower the chance of reaching NEDA-3. A latency value increase of 1 reduced the chance by about 3.4%.

Univariate binary logistic regression showed that the values for the score and latency of the left eye obtained at diagnosis were the only two examined factors that influenced the achievement of NEDA-3. Multivariate binary logistic regression indicated that only the score for the left eye affected the achievement of a favorable treatment outcome.

A normal finding in the left eye affected the achievement of NEDA-3 (r = 0.004, OR = 5850 (1741–19,655)). Patients who had a regular finding were about six times more likely to experience improvements.

Left-sided latency affected the EDSS score (r = 0.041, OR = 1.026 (1.001–1.053)). Thus, higher latency values indicated a greater risk of increased EDSS in the further course of the disease. The use of immunomodulatory therapy indicates a risk of increased EDSS (r = 0.002, OR = 0.111 (0.025–0.486)). The initial value for the EDSS affected its increase (r = 0.002, OR = 1.966 (1.271–3.039)). Namely, the higher the initial EDSS value, the greater the risk of a later increase.

A normal left VEP score indicated a lower risk of increasing EDSS by about four-fold (r = 0.020, OR = 0.262 (0.085–0.812)).

Univariate binary logistic regression showed that the left-hand score influenced the increase in EDSS by four times (*p* = 0.020, OR = 0.262 (0.085–0.812)).

In the examined group of patients with MS, 46 (58.2%) exhibited ON, while 33 (41.8%) never had ON. A total of 24 (52.2%) patients had ON in the left eye, 15 patients (32.6%) had ON in the right eye and 7 patients (15.2%) had ON in both eyes.

The VEP score for the left eye at the beginning of the disease showed a normal finding in 40 (48.2%) subjects, prolonged latency in 18 (21.7%), prolonged latency with reduced amplitude in 23 (27.7%) and an unformed wave in 2 (2.4%). The VEP score for the right eye showed a neat finding in 36 (44.4%) subjects, prolonged latency in 23 (28.4%) and prolonged latency with reduced amplitudes in 22 (27.2%) subjects.

Table 5 shows the correlation of EDSS with latency and amplitude values, and the total VEP score obtained at disease onset.

The current EDSS did not correlate with values for latency in the eye affected by retrobulbar neuritis at the beginning of the disease (r = 0.261, r = 0.087), nor with values for amplitude in the eye affected by retrobulbar neuritis at the beginning of the disease (r =−0.186, r = 0.227).

EDSS was positively correlated with the total VEP score for both eyes at the beginning of the disease (r = 0.308, r = 0.006), the VEP score for the left eye (r = 0.269, r = 0.016), the VEP score for the right eye (r = 0.253, r = 0.025), latency values for the left eye (r = 0.246, r = 0.030) and latency values for the right eye (r = 0.302, r = 0.007).

EDSS values were not correlated with amplitude values for the left eye (r = −0.133, r = 0.245) or for the right eye (r = −0.184, r = 0.107).

## 4. Discussion

The advent of more effective drugs that modify the course of the disease has expanded our choice for the treatment of patients with MS and presents a great challenge for neurologists in deciding on the most appropriate therapy for each individual patient. Everyone agrees that the early initiation of therapy that modifies the natural course of MS is a condition for a favorable outcome with the goal of achieving complete remission. The biggest challenge is to predict the responses to an applied therapy. Early indicators are being sought so that a particular medicament can be replaced by another more effective drug [21].

The expected favorable outcome of treatment NEDA-3 is achieved if the patient shows no worsening of the disease, no progression of physical disability, and no disease activity on the endocranial MRI.

As the need to measure the outcome of disease treatment increases, the percentage of patients who manage to reach and maintain NEDA-3 is used to assess the effectiveness of therapy that modifies its natural course [21]. Among our patients, 18.2% reached NEDA-3 after 10 years of monitoring, 21.1% reached NEDA-3 after 5 years of monitoring and 29% reached NEDA-3 after 2 years of monitoring. In a recent study, Rotstein et al. [14] found that about one-third of their patients attained NEDA-3 after 2 years of monitoring, while about half of the group reached NEDA-3 after 1 year of follow-up.

According to our results, 58.2% of the patients had ON. It was present in 52.2% of patients in the left eye, 32.6% of patients in the right eye and 5.2% of patients in both eyes. One sign of the demyelination process is the finding of extended VEP latencies with preserved wave amplitudes [22].

When compared to MR, the VEP finding shows greater sensitivity in detecting lesions on the optic nerves. Patients exhibiting ON during the course of MS are more likely to have a pathological VEP finding than the detection of noticeable lesions on the optic nerves by MR [23]. However, the importance of VEP in clinical practice is diminished by its lack of sensitivity because prolonged latencies are present in neurosarcoidosis, NMOSD, adrenoleukodystrophy and neurosyphilis [24].

However, the prognostic value of VEP can be assessed using a variety of methods: by predicting the degree of damage to the optic nerve itself and the long-term outcome in terms of visual impairment; by predicting the development of definitive MS in patients with CIS; and by predicting future disability in patients already suffering from MS [2]. Currently, there is increasing discussion about whether VEP has a significant value in predicting a favorable outcome with respect to treatment for patients with RRMS.

Studies conducted in the 1980s and 1990s indicated a significant association between changes in VEP and the subsequent development of MS, pointing to a 2.5 to 9 times increased risk of developing the disease [25,26,27]. However, subsequent investigations showed that the incidence of pathological VEP values did not differ significantly between patients who developed MS and those who did not [28], although the risk of developing the disease increased in patients with VEP findings other than prolonged latencies and amplitude reduction [11,29].

According to our results, the current degree of disability did not correlate with the values for latency and amplitude in the eye affected by ON at the beginning of the disease nor with the amplitudes in the left and right eye. However, the current degree of disability positively correlated with the total VEP score for both eyes at the beginning of the disease, the VEP score for the left and right eye and latency values for the left and right eye regardless of whether patients had ON or not.

The values for the latency and amplitude of VEP in the eye affected by ON at the beginning of the disease were not related to a later higher degree of disability, the frequency of relapse or progression in endocranial MR findings. However, prolonged values of left-sided latency may indicate a higher risk of increased EDSS in the further course of the disease. According to Leocani et al. [2], the long-term outcome of MS cannot be predicted only by VEP obtained in the acute phase of the disease. The persistence of pathological findings in the subsequent months indicated a worse outcome [2]. In a cohort of 28 patients, the median degree of disability measured by EDSS was found to differ significantly between MS patients who had and those who did not have a normal VEP finding [30].

Several studies that addressed the predictive role of VEP on outcomes in terms of physical disability found a moderate correlation between VEP and subsequent physical disability scores [18,31]. Here, we have shown that the initial value of EDSS affects the risk of increasing disabilities. The higher the initial value of EDSS, the greater the risk of a worsening disability in the further course of the disease. Therefore, it is very important to start therapy as soon as possible after diagnosis in order to prevent an increase in physical disability and premature mortality [14].

Patients treated with interferon beta 1a had the highest frequency of EDSS score increases (31.8%), while those given interferon beta 1b had the lowest (6.9%). In our previous clinical, observational, cross-sectional study, we showed a lower median value of EDSS in patients treated with interferon beta 1a in comparison with patients treated with interferon beta 1b [32].

A neat left-sided VEP score indicated a four-fold lower risk of increasing disabilities, and the chance of reaching NEDA increased six-fold. Moreover, VEP scores for the left eye as well as latency values for the left eye at the beginning of patient monitoring affected the achievement of NEDA. If the VEP score was higher by 1, the chance of reaching NEDA was reduced three times; if latency values were extended by 1, the chance of reaching NEDA was about 3.4% less.

Other studies [33] suggested that the topographic location of the lesion, rather than the volume of the lesion, may play a significant role in disability in MS probably due to disconnection mechanisms. The same authors indicated that the localization of the lesion in early stages of MS has a predictive role in clinical exacerbations.

Some Italian authors [34] showed that the development of progressive disability, which was monitored for 5 years, is directly correlated with the development of the irreversible atrophy of white and gray matter in the left hemisphere. They pointed out the importance of the existence of hemisphere lateralization when monitoring clinical disabilities and cognitive impairment. The lateralization of the hemispheres has been confirmed by other authors suggesting different hemisphere susceptibilities to irreversible atrophic changes [35]. Many studies have shown that the hemispheres express different susceptibilities to accumulate damage during disease [36]. In 1995, Filippi pointed out that in right-handed people the dominant hemisphere had a significantly higher load of lesions [37]. The accumulation of damage to the gray matter of the dominant hemisphere has also been observed in Alzheimer’s dementia. Princeter found significantly more damage to the cortical mass in the left hemisphere frontotemporally, especially in the motor areas of patients with RRMS [38].

It is difficult at this time to conclude which factors render one of the hemispheres more susceptible to autoimmune damage. Some authors explain this asymmetric distribution of lesions by greater susceptibility for neural and metabolic dysfunctions in the left hemisphere [39], which may be the reason for more changes in the left optic nerve. The lateralization of the hemispheres also exists in different modulations of immune functions, which has been demonstrated in patients with epilepsy [40]. Thus, numerous studies indicate that optic nerve lesions were closely correlated with white and gray matter atrophy, so ON reflects pathological mechanisms that affect the CNS and lead to axon loss (10). This explains the lateralization of lesions in the optic nerves and the different significance in predicting outcomes for treated patients. Rare research, such as the study by Gabilondo et al., which examined the functional integrity of the posterior visual pathways and the speed of visual processing, revealed damage in MS patients in three regions—the left gyrus lingualis, the left cuneus and the medial part of Brodman’s area 18 [41].

Previous research has not dealt with variabilities in the findings of left- or right-handed VEP, and this is the first study to address this matter.

## 5. Conclusions

The findings for VEP can have a significant predictive value for achieving favorable treatment outcomes in patients with RRMS. The VEP score for the left eye, as well as latency for the left eye at the beginning of the disease in patients receiving first-line immunomodulatory therapy, had predictive values for reaching NEDA-3 over several years.

## Figures and Tables

**Table 1 sensors-22-08849-t001:** Distribution of patients according to the time period of the follow-up visit.

	Male N (%)	Female N (%)	Total N (%)
Ten-year monitoring	11 (13.2)	22 (26.5)	33 (39.7)
Five-year monitoring	8 (9.6)	11 (13.2)	19 (22.9)
Two-year monitoring	11 (13.2)	20 (24.1)	31 (37.3)

**Table 2 sensors-22-08849-t002:** Demographic and clinical characteristics of patients monitored for 10 years, 5 years or 2 years.

	X ± SD	Min–Max	Ten-Year Monitoring	Five-Year Monitoring	Two-Year Monitoring	*p* *
Age of the patients (years)	38.31 ± 9.01	22–58	44.67 ± 8.35	35.32 ± 7.67	33.9 ± 6.11	*p* < 0.0005
Years of education (years)	13.32 ± 3.09	8–24	12.81 ± 3.57	13.37 ± 3.17	13.86 ± 2.41	*p* = 0.432
Treatment duration (months)			89.72 ± 43.09	52.57 ± 25.75	24.35 ± 6.13	*p* < 0.0005
Disease duration (months)	104.81 ± 78.67	18–408	180.75 ± 72.06	72.53 ± 23.96	44.27 ± 18.08	*p* < 0.0005
EDSS	2.11 ± 1.37	0–7.0	2.50 ±0.75	2.0 ± 0.5	2.0 ±1.0	*p* = 0.555

* Kruskal–Wallis test.

**Table 3 sensors-22-08849-t003:** Distribution of NEDA-3 achievement in patients monitored for 2 years, 5 years or 10 years.

	Ten-Year Monitoring	Five-Year Monitoring	Two-Year Monitoring
Achieved NEDA-3 N (%)	6 (18.2)	4 (21.1)	9 (29)
Did not achieve NEDA-3 N (%)	27 (81.8)	15 (78.9)	22 (71)

**Table 4 sensors-22-08849-t004:** Examined factors for achieving NEDA-3 and increase in EDSS.

Achieving NEDA-3	Odds Ratio (95% CI)	*p* *
Optic neuritis	1.542 (0.536–4.437)	0.422
Normal finding on the left eye	5.850 (1.741–19.655)	0.004
Total score for both eyes	0.698 (0.473–1.030)	0.07
Score for the left eye	0.344 (0.156–0.757)	0.008
Latency value for the left eye	0.966 (0.934–0.999)	0.042
Amplitude value for the left eye	1.135 (0.959–1.343)	0.142
Score for the right eye	1.012 (0.538–1.903)	0.971
Latency value for the right eye	1.002 (0.981–1.02)	0.848
Amplitude value for the right eye	1.143 (0.968–1.350)	0.114
Length of treatment	0.996 (0.984–1.009)	0.551
Disease duration	1.004 (0.996–1.012)	0.305
EDSS	1.520 (0.873–2.647)	0.139
**Increase in EDSS**	**Odds ratio (95% CI)**	***p* ***
Latency value for the left eye	1.026 (1.001–1.053)	0.041
Immunomodulatory therapy	0.111 (0.025–0.486)	0.002
Initial EDSS value	1.966 (1.271–3.039)	0.002
Normal finding on the left eye	1.966 (1.271–3.039)	0.020

***** Binary logistic regression.

**Table 5 sensors-22-08849-t005:** Correlation of EDSS with latency and amplitude values and total VEP score obtained at disease onset.

	r	*p* *
Latency for the eye affected by ON	0.261	0.087
Amplitude for the eye affected by ON	−0.186	0.227
Total VEP score for both eyes	0.308	0.006
VEP score for the left eye	0.269	0.016
VEP score for the right eye	0.253	0.025
Latency value for the left eye	0.246	0.03
Latency value for the right eye	0.302	0.007
Amplitude value for the left eye	−0.133	0.245
Amplitude value for the right eye	−0.184	0.107

***** Spearman’s Rank correlation coefficient.

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
