# Peer review of "VEP Score of a Left Eye Had Predictive Values for Achieving NEDA-3 over Ten Years in Patients with Multiple Sclerosis"

_sensors, 2022, doi:10.3390/s22228849_

Round 1
Reviewer 1 Report
The work is quite interesting and does advance knowledge in the area. The paper can, however, be improved in several ways. The English language must be reviewed, beginning with the title. The statistical procedures need to be broadly described beyond referencing the software used since research must not depend on a particular commercial tool - a general description of the statistics used will greatly improve the presentation.
Reviewer 2 Report
The paper needs some editorial corrections (Affiliation 1 is not necessary-authors are from the same institution, bold is not required for the corresponding author), letter size should be kept the same.
The Abstract is modest, Methods subsection should be supplemented on the patient's data on their number (move the sentence from the Results beginning to this part), sex, and age on average; more data on the optical neuritis and disability evaluations would be invited. It should be shortly mentioned, what has shown the MR imaging of the endocranium?
Consider …” Achieving NEDA-3 has been considered as a favorable outcome of treatment.”… instead…”Achieving NEDA-3 is a favorable outcome of treatment.”… (sounds like a result) in M&M section.
In the Results subsection of the Abstract appear EP and OR shorts (Evoked Potential score?) not explained previously.
The sentence …”The EP score (r = 0.008, OR = 0.344 (0.156–0.757)) and latency (r = 0.042, OR = 0.966 (0.934–0.999)) at the onset of the disease were predictive factors for achieving NEDA-3.”… looks like separated from the context.
The introduction is impressive and accurate.
M&M section
Line 70 - a history of corticosteroid application,
Line 77 - All patients underwent VEP (p100) and the amplitude and latency of the waves were measured. – the description of VEP methodology is very modest and needs technical descriptions (amplifications, time base, filters, patients position and distance from the stimulation monitor, what pattern of stimulation, frequency etc.) or refs. describing the principles with previous experiences is required.
The software in line 95 … the SPSS statistical program”… sounds mysterious – producer, country.
Results
The Results are convincing.
Table 2 is confusing in the middle.
This is not obligatory but comparative, most representative VEPs recordings examples will be cordially invited. Data in tables are clear but minor communicative.
Discussion and Conclusions are satisfactory and convincing.
Refs
Line 317 : Neuro-Ophthalmology instead Neuro-ophthalmology
In the lines below dots are sometimes lacking in the shorts of journals’ names
Neuroepidemiology - line 349 – Italic
A help from the English native speaker is required for revision of the minor grammar mistakes.
The paper sounds.
